# Photoredox-catalyzed oxo-amination of aryl cyclopropanes

Liang Ge[1], Ding-Xing Wang[1], Renyi Xing[1], Di Ma[1], Patrick J. Walsh [2]* & Chao Feng [1]*

Cyclopropanes represent a class of versatile building blocks in modern organic synthesis. While the release of ring strain offers a thermodynamic driving force, the control of selectivity for C–C bond cleavage and the subsequent regiochemistry of the functionalization remains difficult, especially for unactivated cyclopropanes. Here we report a photoredox-coupled ring-opening oxo-amination of electronically unbiased cyclopropanes, which enables the expedient construction of a host of structurally diverse $\beta$-amino ketone derivatives. Through one electron oxidation, the relatively inert aryl cyclopropanes are readily converted into reactive radical cation intermediates, which in turn participate in the ensuing ring-opening functionalizations. Based on mechanistic studies, the present oxo-amination is proposed to proceed through an $S_N2$-like nucleophilic attack/ring-opening manifold. This protocol features wide substrate scope, mild reaction conditions, and use of dioxygen as an oxidant both for catalyst regeneration and oxygen-incorporation. Moreover, a one-pot formal aminoacylation of olefins is described through a sequential cyclopropanation/oxo-amination.

[1] Institute of Advanced Synthesis (IAS), School of Chemistry and Molecular Engineering, Nanjing Tech University, 30 South Puzhu Road, 211816 Nanjing, P. R. China. [2] Roy and Diana Vagelos Laboratories, Department of Chemistry, University of Pennsylvania, 231 South 34th Street, Philadelphia, PA 19104, USA. *email: pwalsh@sas.upenn.edu; iamcfeng@njtech.edu.cn

**C**yclopropanes are versatile building blocks in modern organic synthesis, because their unique bonding and high ring strain confers diverse possibilities for ring-opening and elaboration[1–3]. Although the release of ring strain offers a thermodynamic driving force, the control of selectivity for C–C bond cleavage and the subsequent regiochemistry of the functionalization remains difficult. To meet this challenge, a variety of strategies have been devised (Fig. 1a). Among these, heavily functionalized donor–acceptor cyclopropanes (DACs) constitute a privileged class and are predisposed to both facile and selective ring-opening reactions, particularly under Lewis acid catalysis[4–9]. In contrast, the ring-opening functionalization of electronically unbiased cyclopropanes is much more difficult, especially in a regioselective fashion. In this respect, transition-metal catalysis enables ring-opening through either oxidative addition or β-carbon elimination manifolds[10–13]. While promising, specific directing groups or chelating functionalities are usually essential for controlling site-selectivity in metal promoted C–C bond cleavages. Another strategy to activate cyclopropanes takes advantage of their alkene-like reactivity, enabling electrophilic activation with Lewis acidic species. While efficacious protocols in this area have been recently disclosed[14–19], the scope of the nucleophilic components remains limited. Notwithstanding the advances outlined above, the development of protocols for the selective ring-opening and functionalization of non-activated cyclopropanes is a problem that is largely unsolved.

During the past decade, visible-light-promoted photoredox catalysis has emerged as a powerful technique for enabling otherwise difficult synthetic transformations[20–26]. In 2014, Nicewicz and co-workers reported seminal work on intermolecular anti-Markovnikov hydroamination of alkenes by photoredox catalysis, wherein an alkene radical cation intermediate is generated upon single electron oxidation and then undergoes nucleophilic addition by amine derivatives (Fig. 1b)[27]. Based on this work, we employed single electron oxidation of gem-difluoroalkenes to accomplish the fluoroallylation reaction[28]. With these precedents in mind, we hypothesized that the selective

ring-opening/functionalization of aryl cyclopropanes might be possible through photoredox catalysis. As background, high-energy light-driven or strong oxidant-mediated ring-opening functionalization of unactivated cyclopropanes through radical cation intermediates have been reported[29–36], and photoredox functionalizations of electronically activated cyclopropanes are also known[37–49]. In contrast, the visible-light photoredox catalysis approach for synthetic elaboration of relatively unreactive cyclopropanes is uncharted (Fig. 1c). We rationalize that the photoredox-enabled single electron oxidation of such molecules would provide access to transient aryl radical cation intermediates under mild conditions[50–61]. The resulting aryl radical activates the adjacent cyclopropane motif towards intermolecular nucleophilic attack, providing a path for C–C bond cleavage and functionalization.

With our continuing interest in developing original synthetic transformations using photoredox catalysis, we present herein the example of a photocatalytic ring-opening 1,3-oxo-amination of unactivated cyclopropanes using molecular oxygen and azaarene nucleophiles (Fig. 1c)[28,62,63]. Notable features of this approach include: (1) aryl cyclopropanes are readily oxidized via SET to aryl radical cations, which activate the otherwise relatively inert neighboring cyclopropane motif; (2) dioxygen has a dual role in this transformation: it acts as an oxidant for catalyst regeneration and serves to oxygenate the benzylic radical intermediate; (3) the products are β-azaaryl ketones, which are core structural motifs in naturally occurring molecules and also represent key building blocks in organic synthesis[64,65]; and (4) a one-pot formal aminoacylation of simple alkenes is accomplished through a cascade cyclopropanation/oxo-amination (Fig. 1d)[66,67].

## Results

**Reaction optimization.** To probe our hypothesis, cyclic voltammetry measurements of aryl cyclopropane substrate **1a** were conducted, which indicated the oxidative potential of **1a** being around 1.30 V (vs. saturated calomel electrode in MeCN). Based on this measurement, the model reaction between **1a** and pyrazole **2a** was attempted under air in the presence of selected photocatalysts with relatively high oxidation potentials (**PC-I** to **PC-VII**, Table 1). After extensive investigation of reaction parameters (selected examples listed in Table 1), we were pleased to find that using Acr+-Mes (**PC-I**, $E_0^{1/2} = +2.06$ V vs. SCE in MeCN)[68] as the photocatalyst and blue LEDs as the light source, the model reaction in DCE gave the desired β-amino ketone **3a** in 64% yield (Table 1, entry 1). Subsequent examination of various photocatalysts led to [Ir(dF(CF₃)ppy)₂(4,4′-dCF₃bpy)](PF₆) (**PC-III**, $E_0^{1/2}$III*/II = +1.65 V vs. SCE in MeCN)[69] as the best choice (Table 1, entries 2–6). Further fine tuning of the reaction conditions, including the replacement of the air atmosphere with dioxygen, lowering the catalyst loading and adding 4 Å molecular sieves (Table 1, entries 7–9), eventually led to the formation of product **3a** in 85% yield (see Supplementary Tables 1–8). Notably, control experiments clearly demonstrated the need for both the photoredox catalyst and light irradiation (Table 1, entries 10 and 11).

**Oxo-amination of aryl cyclopropanes.** With the optimized reaction conditions in hand (Table 1, entry 9), the reaction scope of cyclopropane substrates was investigated using pyrazole **2a** as the azaarene reaction partner. As presented in Table 2, a variety of aryl cyclopropanes bearing gem-dimethyl substituents were initially examined. Among these, cyclopropanes possessing electron-donating groups on the para-position of the phenyl ring, such as methoxy (**3aa**), alkyl (**3ca**, **3da**), and 4-Ph (**3ah**), reacted smoothly and provided the corresponding β-amino ketones in

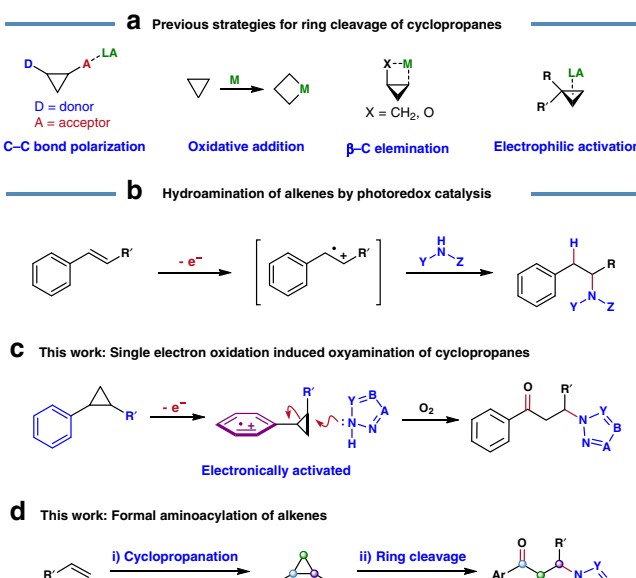

**Fig. 1** Synthetic elaboration of cyclopropanes and alkenes. **a** Reported strategies for ring-opening functionalization of cyclopropanes. **b** Hydroamination of alkenes by photoredox catalysis. **c** Oxo-amination of unactivated aryl cyclopropanes. **d** One-pot aminoacylation of alkenes

**Table 1 Reaction conditions optimization[a]**

| Entry | Catalyst | Yield (%) |
|---|---|---|
| 1 | PC-I | 64 |
| 2 | PC-II | 0 |
| 3 | PC-III | 68 |
| 4 | PC-IV | 65 |
| 5 | PC-V | 47 |
| 6 | PC-VI | 57 |
| 7 | PC-III | 74[b] |
| 8 | PC-III | 85[b, c] |
| 9 | PC-III | 85[b, c, d] |
| 10 | PC-III | 0[e] |
| 11 | – | 0 |

PC photoredox catalyst, DCE 1,2-dichloroethane
[a]Reactions were performed with **1a** (0.1 mmol), **2a** (0.3 mmol), photocatalyst (2 mol%) in 1,2-dichloroethane (0.5 mL), irradiating with 15 W blue LEDs under an air atmosphere at room temperature for 16 h. Yields were determined by [1]H NMR using 1,1,2,2-tetrachloroethane as internal standard
[b]Under $O_2$ atmosphere
[c]With 4 Å molecular sieves
[d]0.2 mol% photoredox catalyst was used
[e]Either without irradiation or under $N_2$ atmosphere

good yields (65–80%). In the case of 4-bromophenyl substrate **1b**, which exhibits relatively high oxidation potential, oxo-amination proceeded in 80% yield, although a prolonged reaction time (48 h) was required for complete conversion (**3ba**). Interestingly, compared with the reaction of the parent **1a**, the 3-OMe and 2-OMe-derived substrates reacted more slowly (**3fa** and **3ga**, 72% and 62% yield, respectively). In the bis(cyclopropyl)arene **1i**, good selectivity for ring opening/functionalization at the most sub-stituted carbon of the more substituted cyclopropyl group was observed, providing 56% yield of **3ia**. Cyclopropanes with dis-ubstituted aryl motifs were generally good substrates (**1j**–**1o**), affording products in good yields even if one of the substituents was moderately (**3ja**–**3pa**) or strongly electron withdrawing, such as an ester or ketone (**3oa**, **3ta**). Cyclopropanes decorated with fused aromatic substituents engaged in this transformation, albeit with diminished yield (46%, **3qa**). Heterocycles are important structural motifs in medicinal chemistry. We were delighted to find that benzofuran derivatives, among others, afforded the desired products in moderate to good yields (39–74%, **3ra**–**3ta**). Here again, good selectivity was obtained in the opening of the bis (cyclopropane) derivative to give **3sa**.

To test if the *gem*-dimethyl substituents were necessary for the success of this reaction, we employed the monosubstituted cyclopropane, 1-cyclopropyl-4-methoxybenzene. The oxo-amination product **3ua** was generated in 60% yield. Aryl cyclopropanes possessing *gem*-dialkyl groups, such as ethyl and cyclopropyl provided **3va** and **3wa** in 44% and 69% yields, respectively. A series of spirocyclic substrates were also amenable to this reaction, providing the desired cycloalkyl amine derivatives in moderate yields (42–49%), regardless of the size of the spirocycles (**3xa**–**3aaa**). Aryl cyclopropanes bearing a single aliphatic substituent, irrespective of the relative *trans/cis* stereo-chemistry, were fine substrates (**3aba**–**3aja**, 42–65% yield). The tolerance of functional groups, such as ester, amide, silyl ether, bromide, etc., in these substrates indicates the generality of the transformation. One limitation is nearby electron-withdrawing groups. When –$CH_2OAc$ was tethered to the cyclopropane (**1ak** and **1al**), reactions under the standard conditions furnished products **3aka** and **3ala** in 20–22% yield. Interestingly, in the case of **1ak** bearing an OMe *para* to the cyclopropyl group, nucleophilic aromatic substitution resulting in displacement of the 4-OMe by the azaarene was the major side reaction. We

**Table 2 Substrate scope of photoredox-catalyzed oxo-amination of aryl cyclopropanes**

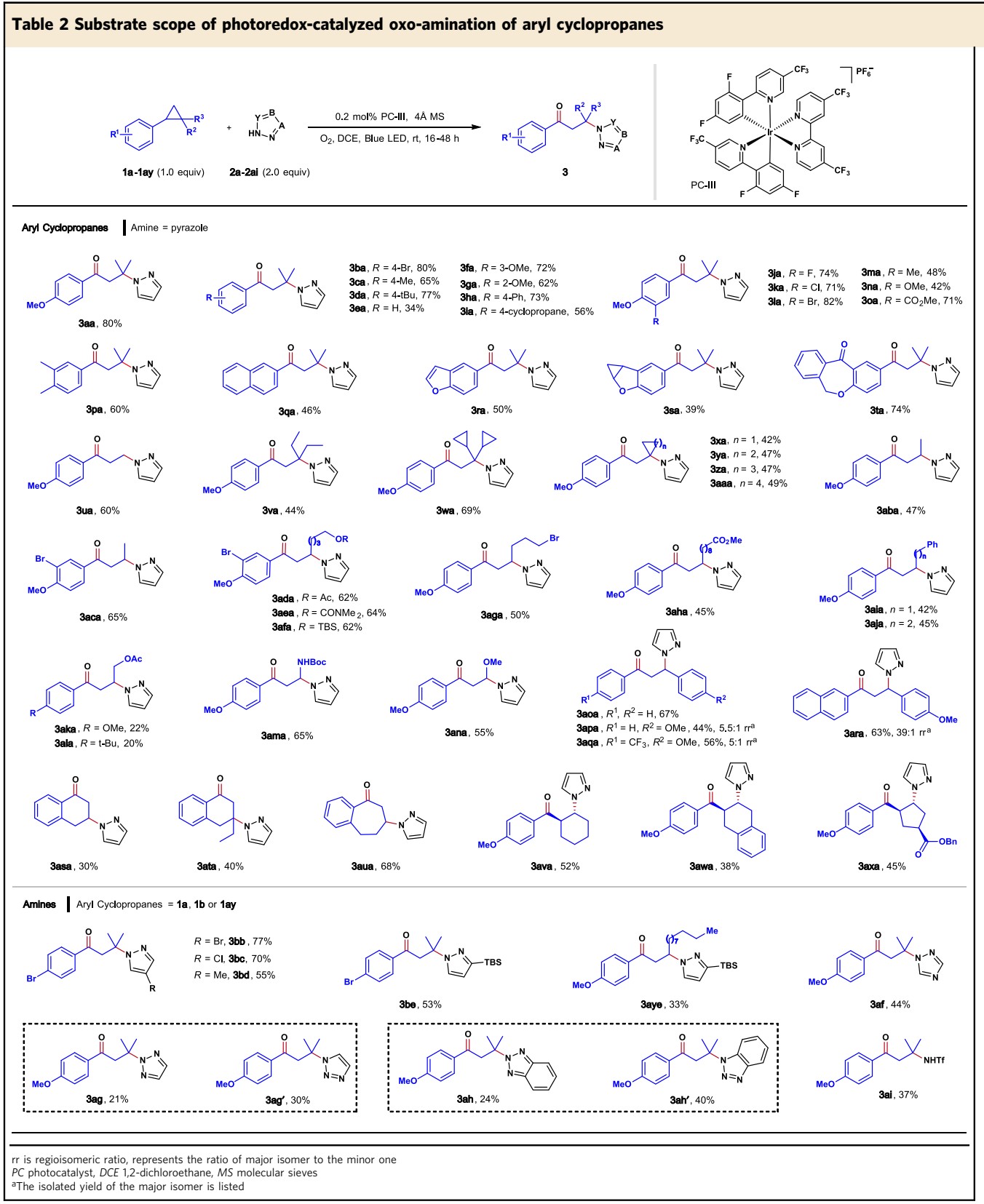

rr is regioisomeric ratio, represents the ratio of major isomer to the minor one
PC photocatalyst, DCE 1,2-dichloroethane, MS molecular sieves
[a]The isolated yield of the major isomer is listed

propose that the inductively electron-withdrawing group (EWG) renders the carbon atom of the cyclopropane less able to participate in stabilization of the arene radical cation and, therefore, less capable of undergoing nucleophilic attack by azaarenes (also see the mechanistic studies below). When cyclopropanes bearing amino or alkoxy substituents were employed, however, the reaction occurred readily to provide a β-oxo amino derivative **3ama**, **3ana** in 65% and 55% yields,

respectively. These substituents can help to stabilize the radical cation intermediate.

We were curious about the utility, and especially the regioselectivity, when using 1,2-diaryl cyclopropanes substrates with two electronically distinct aryl groups. We found that these substrates were indeed viable in the oxo-amination (**3aoa–3ara**). In the case of unsymmetrical 1,2-diaryl cyclopropanes, the regioisomeric ratio (rr) was about 5:1, with the major isomer derived from attack of the nucleophile at the carbon of the cyclopropane bearing the more electron-donating aryl group. We propose that the positive charge of the radical cation intermediate is delocalized over the entire molecule. The attack of pyrazole targets the cyclopropane carbon with the more electron-donating aryl substituent, because the positive charge is more readily accommodated at that position.

Bicyclic cyclopropanes undergo the ring opening/functionalization reactions in 30–68% yield (**3asa–3axa**). It is noteworthy that these reactions furnish only the *trans* diastereomers, as judged by ¹H NMR. This outcome suggests that perhaps the nucleophilic attack proceeds via a concerted mechanism (see the mechanistic studies below for further discussion).

The scope of the *N*-nucleophiles in the oxo-amination was next briefly examined. It was found that pyrazole derivatives with four substituents, such as Br, Cl, and Me, took part in this reaction and afforded the products in moderate to good yields (**3bb–3bd**, 55–77%). When 3-TBS-substituted pyrazole was subjected to the standard reaction conditions, products **3be** and **3aye** were obtained in 33–53% yield, with the amination selectively occurring on the sterically more accessible nitrogen. In addition to pyrazoles, triazole derivatives also participated in this chemistry. While 1,2,4-triazole reacted selectively to deliver the product **3af** in 44%, the use of 1,2,3-triazole derivatives afforded two regioisomeric products (**3ag**, **3ag′** and **3ah**, **3ah′**) with good overall conversion.

It is worth pointing out the prerequisite for the success in this reaction is the efficient and selective single electron oxidation of aryl cyclopropane substrates. The aza-nucleophile cannot, therefore, interfere with this oxidation. Based on the oxidation

potentials of prospective amine nucleophiles (see Supplementary Table 9), trifluoromethanesulfonamide (TfNH₂) was identified as a potential nucleophile. As proof-of-concept, the amination product **3ai** was obtained in 37% unoptimized yield under the standard reaction conditions[27].

**One-pot formal aminoacylation of alkenes**. To streamline the synthesis of the oxo amination process, we wondered if it would be possible to merge the alkene cyclopropanation with our ring-opening functionalization. Fortunately, after considerable effort, we succeeded in integrating Yoshida's cyclopropanation protocol with our ring-opening oxo-amination, thus enabling a direct access to elaborated *β*-amino ketone derivatives from readily available alkene substrates[70]. As shown in Fig. 2, the one-pot aminoacylation proceeded readily with aliphatic alkenes, delivering the desired *β*-amino ketone products in synthetically useful yields. For example, 1-dodecene participated in this cascade transformation to afford the product **3aya** in 39% yield over two steps. Aromatic alkenes, such as styrene and 2-vinylnaphthalene, also participated in this transformation. Here the major products **3apa** and **3ara** were formed by attack of the amine on the carbon bearing the more electron-rich aryl group, consistent with the reactions of unsymmetrical 1,2-diaryl cyclopropanes discussed above. Furthermore, this one-pot procedure accommodated relatively complex alkene substrates. When the isoxepac-derived terminal alkene was employed, the reaction delivered the desired product as a mixture of two regioisomers (**3aza**, **3aza′**) in 69% yield, with nucleophilic attack on the more electron-rich carbon of the cyclopropane predominating. In order to further explore the practicality of this cascade transformation, a gram-scale aminoacylation of 1-dodecene was pursued with a reduced catalyst loading (0.02 mol% **PC-III**). Pleasingly, the desired product **3aya** was isolated in 38% yield over two steps (1.15 g). This result demonstrates the potential of this method for the expedient assembly of a library of structurally diverse *β*-amino ketone derivatives from simple olefins, which

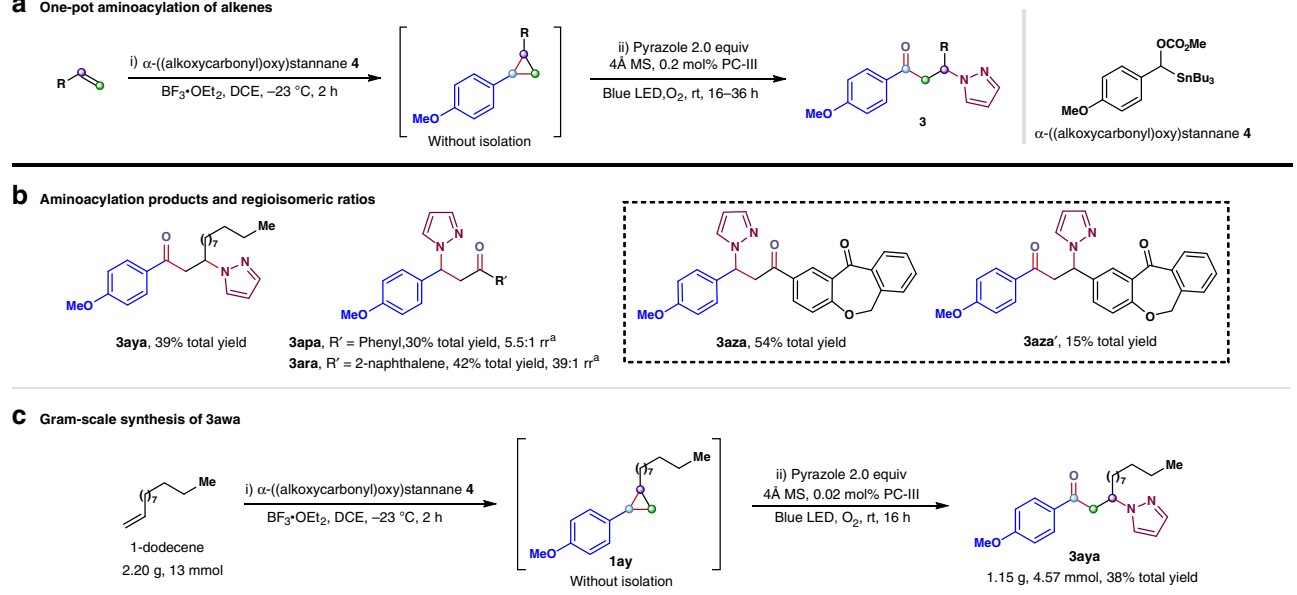

**Fig. 2** Telescoped reactions and scale-up experiments. **a** One-pot formal aminoacylation of alkenes. **b** Additional products and their regioselectivity. **c** Gram-scale reaction to form **3aya**. ªThe isolated yield of the major isomer is listed. rr is regioisomeric ratio, represents the ratio of major isomer to the minor one. PC photoredox catalyst, DCE 1,2-dichloroethane, MS molecular sieves

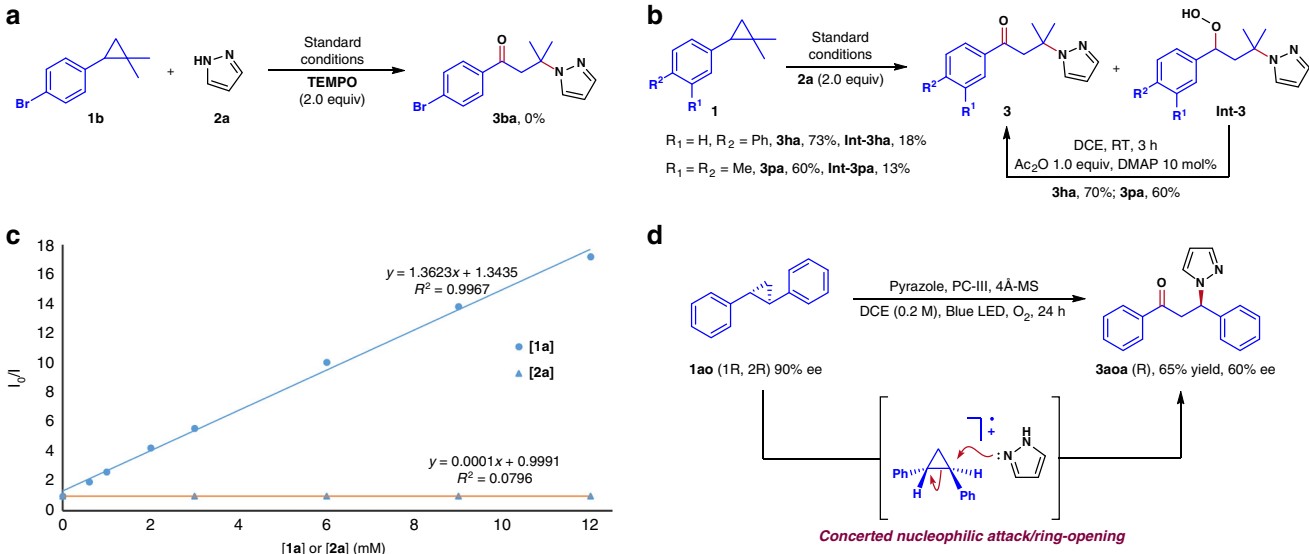

**Fig. 3** Mechanistic investigations. **a** Inhibition by TEMPO of the reaction between **1b** and **2a**. **b** Isolation of hydroperoxide intermediates and their conversion to the products **3**. **c** Stern–Volmer fluorescence quenching study of **1a** and **2a**. **d** Experiments with enantiomerically enriched **1ao**. DCE 1,2-dichloroethane, MS molecular sieves, DMAP 4-N,N-dimethylaminopyridine

could be of importance in pharmaceutical-related research programs.

**Mechanistic studies.** We next set out to study the mechanism of this interesting transformation. First, control experiments conducted with intermittent light illumination indicated continuous irradiation is necessary to maintain conversion. To further probe the reaction pathway the reaction was conducted in the presence of TEMPO. In the event, reaction of **1b** and **2a** in the presence of stoichiometric TEMPO inhibited the oxo amination (Fig. 3a). These results are consistent with a radical mechanism. Furthermore, the reaction between **1a** and **2a** was determined to have a quantum yield of 0.14, discounting the involvement of a radical chain process (see Supplementary Discussion for details).

Insight into the oxo amination was also obtained through isolation of alkyl hydroperoxide intermediates. For example, in the case of **1h** and **1p**, hydroperoxide intermediates **Int-3ha** and **Int-3pa** were isolated, characterized, and transformed to the oxo amination products by treatment with acetic anhydride and catalytic DMAP (Fig. 3b). The hydroperoxides implicate the involvement of benzylic radicals that add to dioxygen[71].

We next focused on the photoredox initiation of the reaction. Stern–Volmer fluorescence quenching experiments were carried out with cyclopropane **1a** and pyrazole **2a**. It was found that only cyclopropane **1a** effectively quenched the excited state photocatalyst **PC-III** (Fig. 3c).

To probe the cyclopropane ring cleavage step, enantiomerically enriched trans-1,2-diphenyl cyclopropane **1ao** (90% ee) was subjected to the standard reaction conditions, which resulted in the generation of product **3aoa** with 60% ee (Fig. 3d). This result indicates that the oxo-amination mainly proceeds through a concerted nucleophilic attack/ring-opening ($S_N2$-like process). The erosion of stereochemistry could result from a minor contribution of ring-opening followed by nucleophilic attack (an $S_N1$-like process). The possibility of racemization of enantiomerically enriched trans-1,2-diphenyl cyclopropane substrate through sensitization via triplet energy transfer, homolytic ring cleavage and radical recombination-based ring closure was also considered. To examine this

possibility, the reaction of (**1R, 2R**)-**1ao** was monitored by chiral phase HPLC. Following the ee as a function of time clearly demonstrated that the ee of both cyclopropane substrate and oxo-amination product degraded with time under the reaction conditions. Furthermore, the diastereoselective transformations of bicyclic substrates **3ava–3axa**, which gave trans-products, are also in accord with a concerted nucleophilic attack/ring-opening manifold. Taken together, we favor the concerted nucleophilic attack/ring-opening mechanism as the predominant pathway.

Based on the experimental results above, and in combination with the cyclic voltammetry experiments described earlier, a reaction mechanism for the oxo-amination of cyclopropanes is proposed (Fig. 4). Initially, excitation of the Ir catalyst generates the excited state $*Ir^{III}$. Reductive quenching by the aryl cyclopropane **1** affords cyclopropyl aryl radical cation intermediate **I** and $Ir^{II}$. At this juncture, nucleophilic attack of pyrazole selectively targets the position better able to accommodate the positive charge to afford radical intermediate **II**. Intermediate **II** is trapped by dioxygen to deliver alkylperoxyl radical **III**. Single electron transfer (SET) between intermediate **III** and $Ir^{II}$ regenerates $Ir^{III}$ with concomitant formation of hydroperoxide **IV**, which is primed to undergo $H_2O$-elimination to provide the product **3**.

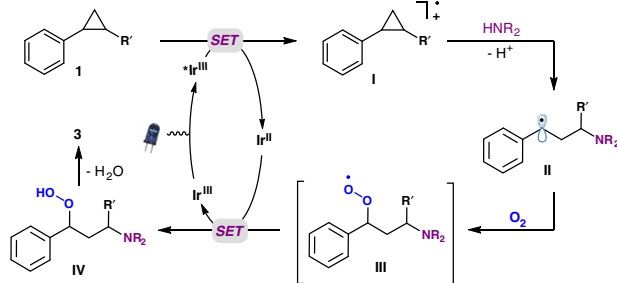

**Fig. 4** Proposed reaction mechanism. Single electron oxidation of aryl cyclopropane **1** gives cation radical intermediate **I** with an activated cyclopropane. Reaction with the nucleophile provides intermediate II, which reacts with dioxygen ultimately affording the product **3**

In the final stages of this manuscript submission, a related work by König and coworkers appeared using aryl cyclopropanes in the presence of HCl, HNO₃, and dioxygen under irradiation to provide 3-chloro-1-phenylpropan-1-one derivatives (ArCOCH₂CH₂-Cl)[42]. Here, a cyclopropyl aryl radical cation (**I**, Fig. 4, R′ = H) was also proposed as the reactive intermediate, formed upon SET from the aryl cyclopropane to •Cl. Nucleophilic attack by chloride followed by reaction with dioxygen furnished the product. Although the products of our reaction and the König chemistry are distinct, and the classes of aryl cyclopropane substrates different (König's works only with the parent cyclopropyl group), these reactions have some common mechanistic intermediates, providing support for the proposed reaction pathway in Fig. 4.

## Discussion

We have developed a photoredox-coupled oxo-amination of electronically unactivated aryl cyclopropanes. The key to success of this transformation hinges upon the selective oxidation of cyclopropyl arene substrates, activating them toward nucleophilic attack by azaaryl nucleophiles. The regioselectivity of attack on cyclopropyl radical cation intermediates is governed by the electronic effects of substituents on the cyclopropane ring. We envision that the strained C–C bond of the cyclopropyl group aligns with the pi-system of the aryl, with the greatest stabilization occurring when the carbon bearing the electron donating alkyl group(s) or electron donating aryl is involved in this interaction. This conformation results in elongation of the cyclopropane C–C bond, increasing the partial positive charge on the carbon bearing the electron donating substituents. The nucleophile then attacks at this carbon, forming the benzylic radical. Based on the results obtained in using enantiomerically enriched 1,2-*trans*-diphenyl cyclopropane (**1ao**) and also the outcomes of diastereoselective transformations of the bicyclic cyclopropanes (**1av**–**1ax**), a concerted nucleophilic attack/ring-opening is favored as the dominant pathway.

Mechanistic and photophysical experiments, isolation of reaction intermediates, and cyclic voltammetry position us to propose a reasonable reaction mechanism (Fig. 4). The key step in the reaction is the initial SET from the aryl cyclopropane to the photoredox catalyst to generate cyclopropyl aryl radical cations under mild visible light conditions. Nucleophilic attack forms a benzylic radical that adds to dioxygen, ultimately generating the carbonyl group.

The reaction introduced herein is noteworthy because it employs readily available azaarenes and molecular oxygen as reagents and it rapidly introduces functionality. Furthermore, the products belong to the family of β-amino ketones, which are a well-known substructures in medicinal chemistry and are valuable building blocks in synthesis. By merging cyclopropanation with the present ring-opening difunctionalization, a one-pot protocol for the aminoacylation of simple alkenes is achieved. Given the abundance of methods to prepare diverse aryl cyclopropanes, we anticipate this method will be applicable to the synthesis of highly functionalized materials. Future efforts will be directed toward expanding the classes of nucleophiles and trapping electrophiles that participate in this difunctionalization reaction.

## Methods

**General procedure for oxo-amination of aryl cyclopropanes**. To an oven dried 10 mL tube equipped with a stirring bar, cyclopropane **1** (0.2 mmol), pyrazole **2a** (0.4 mmol), photocatalyst [Ir(dF(CF₃)ppy)₂(4,4′-bpy)](PF₆) (PC-III) (0.5 mg, 0.2 mol%), 4 Å-MS (100 mg) and anhydrous DCE (1.0 mL, 0.2 M) were sequentially added at room temperature. The reaction tube was capped with rubber septum, charged with an O₂ balloon, and the resulting mixture was irradiated under 15 W blue LED light at room temperature. When the reaction was determined to be completed by TLC, the mixture was passed through a short pad of celite and rinsed with DCM. The filtrate was evaporated to dryness under reduced pressure and the crude residue was purified by column chromatography on silica gel (PE:EtOAc = 9:1–4:1) to afford the desired product **3**.

## Data availability

The authors declare that all the data supporting the findings of this study are available within the paper and its supplementary information files, or from the corresponding author upon request.

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

## Acknowledgements
This paper is dedicated to the 70th anniversary of the Dalian Institute of Chemical Physics, CAS. We gratefully acknowledge the financial support of the "Thousand Talents Plan" Youth Program, the "Jiangsu Specially-Appointed Professor Plan", the "Innovation & Entrepreneurship Talents Plan", the National Natural Science Foundation of China (21871138), and the Natural Science Foundation of Jiangsu Province (BK20170984). P.J. W. thanks the US National Science Foundation for generous support (CHE-1464744 and CHE-1902509).

## Author contributions
L.G. performed most of the experiments and mechanistic study. D.-X.W., R.X. and D.M. took part in the preparation of catalysts and cyclopropane substrates. P.J.W. discussed the chemistry, suggested experiments and revised the manuscript. C.F. conceived the study, directed the project and wrote the manuscript with the assistance of L.G.

## Competing interests
The authors declare no competing interests.
