## [Peer Review File · Nature Communications]

Reviewers' comments:

Reviewer #1 (Remarks to the Author):

Although direct functionalization of alkenes via single electron oxidation by photoredox catalysis has been well explored in recent years since Nicewicz's seminal work, visible-light promoted ring-opening functionalization of cyclopropanes is rare. In this manuscript, Walsh and Feng report a mild and efficient ring-opening oxo-amination of aryl cyclopropanes with the aid of visible-light. This is a conceptually new approach of difunctionalization, especially oxo-amination, of aryl cyclopropanes. The reaction shows broad substrate scopes. A range of aryl and bicyclic cyclopropanes are well tolerated to provide the desired product in fairly good yields. The manuscript is well written and the supporting information provides all the expected data. In conclusion, I recommend acceptance of this work for publication in Nature Communications after addressing some minor points.

1) the structures of 3apa in Table 2 and Scheme 2 are not consistent, please correct or clarify.

2) concerning the regioselectivity of diaryl cyclopropanes, the authors should comment on the ratio of 3axa and 3axa', as it is contradictory to the supposed mechanism. The possibility of ring-opening of diaryl cyclopropane radical cation before nucleophilic attack of azaarenes should be considered or discussed. For this reason, it should be useful to show if enantiopure diaryl cyclopropanes, such as 1,2-diphenylcyclopropane, could be transformed to the product with or without loss of stereochemical information.

3) In the section of "One-pot formal aminoacylation of alkenes", the product of the gram-scale experiment should be "3awa", instead of "3axa"

Reviewer #2 (Remarks to the Author):

Feng and Walsh describe a new photoredox-catalyzed method to convert aryl cyclopropanes into β -amino ketones by ring opening. Although conceptually one could imagine the same products being made by 1,4-addition of the heterocycle to an enone, heterocyclic β -amino ketones obtained by this method have not been described yet in literature. The method could be also interesting for medicinal chemistry to access new structures. The only problem in this regard could be the necessary removal of remaining tin compounds from the cyclopropanation step. A broad range of different substrates was investigated and also a one-pot procedure including the cyclopropanation step is described.

The substrate scope is broad and many new compounds are described within this manuscript. The yields of the reactions are most of the time in a synthetically useful range. Also experiments to get an insight into the mechanism of the reaction have been performed. The low catalyst loading and the use of oxygen for catalyst regeneration and as a reagent to form the intermediate hydroperoxide make the protocol attractive.

The method seems to be limited to heterocyclic amines but this still allows access to many different products using the described procedure. Some parts of the manuscript should be reworked to make it more readable and understandable (highlighted in yellow in the attached manuscript pdf file).

The supporting information is detailed and contains the information needed to reproduce the experiments. Most of the NMR spectra shown are of sufficient purity. There are only some exceptions (for example for compounds **1q** page S68, **30a** page S105 impurities are visible). In these cases, the authors should provide cleaner spectra. Also, for solid products melting points should be provided.

The paper should be accepted after including the indicated minor revisions below:

Suggestions to Improve the Manuscript:

Please delete all the emotional qualifiers (for example beautiful, formidable..)

Page 1: "A photoredox-coupled ring-opening oxo-amination of electronically unbiased cyclopropanes is presented, which enables an expedient construction of a host of structurally diverse β -amino ketone derivatives. By taking advantage of a one electron oxidation, the otherwise recalcitrant donor-donor cyclopropanes are readily converted into reactive donor-acceptor cyclopropanes, which in turn readily participate in the ensuing ring-opening functionalizations." The first and second sentence of the abstract are difficult to understand and many uncommon words were used that will make it challenging for non-native readers. The authors should rephrase these sentences.

In the group of references 14-18, the following relevant reference should be added: *Chem. Sci.* **2018**, *9*, 6411

What is the context of references 36-48? It was unclear from the context of the sentence in which the references appeared.

Page 2: "ring-opening and elaboration" ring-expansion?

Page 2: "Although the release of ring strain has been leveraged for functionalization reactions, a ~~pivotal~~ **pivotal** challenge is to control the selectivity of C–C bond cleavage reactions and the subsequent regiochemistry of the functionalization." Might be difficult to understand

Page 2: " In stark contrast, electronically unbiased cyclopropanes prove recalcitrant toward ring-opening, much less in a regioselective fashion" Might be difficult to understand

Page 2, Scheme 1: The symbol for "electron" under b) and c) should be changed from e to e⁻

Page 3, Table 1: "0.2 mol% PC" should be changed to "2.0 mol% PC" because these are the standard conditions used for the optimization

Page 3: "~~derivatives~~ **derivatives**"

Page 3: "~~β-oxo aminal~~ **amino derivative 3ala**"

Page 4, Table 2: The general structure of **1** should bear also R³ at the same carbon that R² is connected in analogy also for the product **3**. For better understandability "**2**" should be called "**2a–2h**" and in the scheme instead of "**1**" there should be written "**1a–1ax**"

Page 4, Fig. 1: The figure has four parts (a, b, c, d) but the description only shows a and b. This way the figure is not totally understandable. For part b the yields of **3** and **Int-3** should be added and also the yield of the conversion from **Int-3** to **3**.

Page 6, Scheme 3: For the structure of **I** it might help if one of the electrons of the drawn structure is actually removed to make it more clear, that it is a radical and positively charged, even though the way it is drawn at the moment is not wrong. For the step from **I** to **II** "-H⁺" should be added below the arrow. Has the influence of additional base been investigated or is this the reason why an excess of the heterocyclic amine is used? At the step from **IV** to **3** "-H₂O" should be added beneath the arrow to make clear that it's a dehydration reaction. Product **IV** should use the same color code for the NR₂-group like before (red)

Page 6: "nucle**o**philes"

Page 7: "~~voltametry~~ **voltammetry**"

Supporting information:

Page S22: Table S3: "Bule-LEDs" should be changed to "Blue-LEDs"

Page S33ff: Often the frequency shown for the ^1H NMR is 401 MHz and the ^{13}C NMR 101 MHz but later in the shown spectra it's always 400/100 MHz. The authors should make this consistent.

For compounds **1q** page S68, **30a** page S105 impurities are visible, cleaner spectra should be provided

Reviewer #3 (Remarks to the Author):

The manuscript by Walsh, Feng and co-workers details a novel method for the synthesis of beta-aminoketones from aryl cyclopropanes via a photoredox-catalysed cyclopropane ring-opening. The reaction proceeds via single electron oxidation to generate cyclopropane-substituted aryl radical cations, which undergo ring opening/nucleophilic attack by aza-heterocycles to form benzylic radicals. Subsequent reaction with molecular oxygen delivers the aryl ketone product.

This work nicely builds on established reactivity of aryl cyclopropanes, which are known to undergo single electron oxidation/nucleophilic attack/cyclopropane ring-opening to give benzylic radicals (see references 28–31, 33–35). Such reactions have previously only been performed with stoichiometric strong oxidants or with UV light sensitisation. In the present work, the authors demonstrate that the same radical intermediates can be accessed using mild photoredox conditions. Previous photoredox-catalyzed cyclopropane ring openings have been limited to easily oxidisable aminocyclopropanes and only applied to [3+2] cycloadditions with olefins (references 36–40). However, taking inspiration from the anti-Markovnikov hydroaminations of styrenes reported by Nicewicz (reference 26), the authors recognised that arylcyclopropanes could be oxidised by highly oxidising photocatalysts. The resulting radical cation intermediates show distinct reactivity from those generated from aminocyclopropanes, so could be intercepted in oxo-amination reactions to provide useful beta-amino ketone products.

The reaction proceeds under mild and operationally simple conditions using easily accessible starting materials. An extensive substrate scope is demonstrated with respect to the cyclopropane, and several aza-heterocycle nucleophiles are used, effectively highlighting the generality of the methodology. Particularly attractive is the one-pot alkene amino acylation procedure. Although the yields are low, this transformation would not be possible in such a concise fashion using other methods. As a result, I think this a very interesting method that will likely be of interest to the readers of Nature Communications. I recommend publication after addressing the following points:

Manuscript:

- 1) A reference to Scheme 1d should be added to the main text.
- 2) In Table 1, the conditions above the reaction arrow states 0.2 mol% photocatalyst. Either this should be changed to 2.0 mol% or footnote d needs to be changed.
- 3) The regioselectivity, with respect to the nucleophilic attack appears to occur through an SN-1-type process, with attack at the more substituted carbon. This would imply that reaction should be promoted by groups able to stabilise a carbocation. Therefore, the selectivity in product 3sa is intriguing as cyclopropane opening on the furan moiety would be promoted by oxocarbenium stabilisation. Is there an explanation for the observed regioselectivity? As the yield is low, is it possible that the other regioisomer is formed but the amination product is unstable?
- 4) For products 3aja and 3aka in Table 2, it is stated that the low yield is a result of an "unfavourable inductive effect of the -CHOAc on the proximal radical cation intermediate". This is not clear. Is it implied that the radical cation is formed and then destabilised? As this would surely help the nucleophilic attack. Or is it implied that the radical cation is not generated in the first place, due to a more difficult oxidation? If the remaining mass balance is unreacted starting material, it would suggest the latter. I was wondering whether the low yield could be a result of competing intramolecular attack of the radical cation intermediate by the carbonyl oxygen of the acyl group (for similar process, see Chem. Sci. 2015, 6, 270). Some clarification is needed.
- 5) For products 3ana–3apa, regioisomeric ratios should be included.
- 6) What are the diastereomeric ratios of products 3ata, 3aua and 3ava in Table 2? These values need to be included.
- 7) The statement "It is also worth pointing out that the azaaryl nucleophiles amenable to this reaction are, at the present stage, azaarenes with relatively high oxidation potentials" should be elaborated on or rephrased. What does "relatively high oxidation potential" mean? Relative to

what? Presumably, this is to prevent competing single electron oxidation of the nitrogen nucleophile. With respect to this, have the limits of this reaction been examined? Are there examples of nucleophiles with lower oxidation potentials that have failed to react? A nice addition would be to include a brief survey of amine nucleophiles whose oxidation potentials are known/measured by CV and compared to that of the cyclopropanes. If the oxidation potential of the amine nucleophile is lower than that of the cyclopropane, does the reaction fail?

8) Did the authors try triflamide as the aza-nucleophile? This was used by Nicewicz (ref. 26) and provides the potential for subsequent deprotection to access primary amine products.

9) The title for Figure 1 is not correct. It is missing descriptions for a and the labels for b–d are wrong.

10) In figure 1b, the yields for the conversion of Int-3ha and Int-3pa into 3ha and 3pa should be given.

11) The Stern-Volmer quenching studies should also be carried out with molecular oxygen, to rule out an oxidative quenching pathway.

12) For ease of comparison, I think the Stern-Volmer plots would be better presented on a single graph.

13) A reference is needed for König's work (ref. 44) during the discussion of the mechanism in scheme 3.

14) In the discussion it is stated "Attack occurs at the more substituted carbon of the cyclopropyl radical cation, likely due to the greater partial positive charge at this position. Consistent with this proposal, under standard conditions the reaction of the unsymmetric trans-1,2-diarylcyclopropane bearing both a 4-OMe substituent on one aryl and a 4-CF₃ on the other results in selective formation of the keto group adjacent to the electron donating 4-C₆H₄-OMe group." These two regioselectivity arguments are not related. The first relates to the regioselectivity of the cyclopropane ring opening (quaternary position vs. secondary position) and the second relates to regioselectivity of single electron oxidation (electron-rich aromatic vs. electron deficient aromatic).

Supporting Information:

1) Add a Table of contents.

2) The diastereomeric ratios of products 3ata, 3aua and 3ava need to be given. The NMR spectra suggest a single diastereomer. The major diastereomer should be assigned and the relative stereochemistry included in both the supporting information and the manuscript.

3) For the photon flux measurement, two different values are given: 7.41×10^{-7} and 4.27×10^{-7} einstein/s. Which is correct?

4) The photon flux calculation should be repeated with more time points at less than 90 s to check the accuracy of the value. Based on the numbers given, 4.27×10^{-7} moles of Fe²⁺ are generated, however, only 0.35 mL of a 5.55 mM phenanthroline solution is added, which allows for a maximum of only 6.48×10^{-7} moles of Fe(phen)₃²⁺. Therefore, it is likely that all the phenanthroline has been consumed, which will result an incorrect photon flux value.

5) There is a problem with the references. E.g. for the quantum yield measurements, references 13–17 are given, however, only 15 references are given in the Supporting Information.

6) The NMR of products 3ada (p. S121), 3aea (p. S122), 3ara (p. S135) and 3asa (p. S136) have significant impurities present. The products should be re-purified, and the yield recalculated.

Reviewer: Adam Noble

Reviewer #1 (Remarks to the Author): Although direct functionalization of alkenes via single electron oxidation by photoredox catalysis has been well explored in recent years since Nicewicz's seminal work, visible-light promoted ring-opening functionalization of cyclopropanes is rare. In this manuscript, Walsh and Feng report a mild and efficient ring-opening oxo-amination of aryl cyclopropanes with the aid of visible-light. This is a conceptually new approach of difunctionalization, especially oxo-amination, of aryl cyclopropanes. The reaction shows broad substrate scopes. A range of aryl and bicyclic cyclopropanes are well tolerated to provide the desired product in fairly good yields. The manuscript is well written and the supporting information provides all the expected data. In conclusion, I recommend acceptance of this work for publication in Nature Communications after addressing some minor points.

Response: We thank the reviewer for their supportive comments.

Reviewer 1: 1) the structures of **3apa** in Table 2 and Scheme 2 are not consistent, please correct or clarify.

Response: The structure of **3apa** from the original submission (changed to **3ara** in the present manuscript) has been confirmed. This mistake has been revised, see Table 2 and Scheme 2.

Reviewer 1: 2) concerning the regioselectivity of diaryl cyclopropanes, the authors should comment on the ratio of **3axa** and **3axa'**, as it is contradictory to the supposed mechanism. The possibility of ring-opening of diaryl cyclopropane radical cation before nucleophilic

attack of azaarenes should be considered or discussed. For this reason, it should be useful to show if enantiopure diaryl cyclopropanes, such as 1,2-diphenylcyclopropane, could be transformed to the product with or without loss of stereochemical information.

Response: Based on the reviewer's comments on the ring-opening functionalization of unsymmetrical diaryl cyclopropanes, the structures of **3ana** (changed to **3apa**) and **3aoa** (changed to **3aqa**) in Table 2 were revised based on ^1H NMR of each pair of regioisomers, and also comparison with the authentic samples (**3apa**, **3apa'**, **3aqa**, **3aqa'**) made using Aza-Michael type additions.

One-pot aminoacylation of alkenes

We would like to thank the referee for their suggestion of examining the enantioenriched *trans*-1,2-diphenylcyclopropane. We have conducted this reaction using enantioenriched diphenyl cyclopropane (**1R,2R**)-**1am** (changed to (**1R,2R**)-**1ao**). It was found that the reaction of **1am** (changed to **1ao**) of 90% ee led to the generation of the product **3ama** (changed to **3aoa**) in 65% yield with 60% ee after 24 h. This result indicates that the oxoamination mainly proceeds through a concerted nucleophilic attack/ring-opening (S_N2 -like process). The erosion of stereochemistry could arise from a minor contribution of ring-opening followed by nucleophilic attack (S_N1 -like process). However, the racemization of enantiomerically enriched diphenyl cyclopropane substrate **3ama** through sensitization via triplet energy transfer, homolytic ring cleavage and radical recombination-based ring closure was also considered. To examine this possibility, the reaction of (**1R,2R**)-**1am** (changed to (**1R,2R**)-**1ao**) was monitored by chiral phase HPLC. Following the starting material as a function of time clearly demonstrated that the ee of the cyclopropane substrate degraded with time under the reaction conditions. Furthermore, the diastereoselective transformations of bicyclic cyclopropanes to **3ata-3ava** (changed to **3ava-3axa**) are also in accordance with concerted nucleophilic attack/ring-opening. Taken together, we favor the concerted ring-opening mechanism as the predominant pathway. The experimental results have been added in the main text (page 6, right column, lines 16-37) and related experimental details have been added to the SI (pages 43-50).

Control experiment with enantiomerically enriched diphenyl cyclopropane

The evolution of ee values of substrate (1R,2R)-1ao and product (R)-3aoa as the reaction progresses

Diastereoselective ring-opening oxo-amination of bicyclic substrates

Reviewer: 3) In the section of “One-pot formal aminoacylation of alkenes”, the product of the gram-scale experiment should be “**3awa**”, instead of “**3axa**”

Response: This typo has been corrected.

Referee 2

Reviewer 2: Feng and Walsh describe a new photoredox-catalyzed method to convert aryl cyclopropanes into β -amino ketones by ring opening. Although conceptually one could imagine the same products being made by 1,4-addition of the heterocycle to an enone, heterocyclic β -amino ketones obtained by this method have not been described yet in literature. The method could be also interesting for medicinal chemistry to access new structures. The only problem in this regard could be the necessary removal of remaining tin compounds from the cyclopropanation step. A broad range of different substrates was investigated and also a onepot procedure including the cyclopropanation step is described. The substrate scope is broad and many new compounds are described within this manuscript. The yields of the reactions are most of the time in a synthetically useful range. Also experiments to get an insight into the mechanism of the reaction have been performed. The low catalyst loading and the use of oxygen for catalyst regeneration and as a reagent to form the intermediate hydroperoxide make the protocol attractive. The method seems to be limited to heterocyclic amines but this still allows access to many different products using the described procedure. Some parts of the manuscript should be reworked to make it more readable and understandable (highlighted in yellow in the attached manuscript pdf file).

The supporting information is detailed and contains the information needed to reproduce the experiments. Most of the NMR spectra shown are of sufficient purity. There are only some exceptions (for example for compounds 1q page S68, 30a page S105 impurities are visible). In these cases, the authors should provide cleaner spectra. Also, for solid products melting points should be provided. The paper should be accepted after including the indicated minor revisions below:

Response: We thank the reviewer for their supportive comments.

Reviewer 2: 1) Please delete all the emotional qualifiers (for example beautiful, formadible..)

Page 1: "A photoredox-coupled ring-opening oxo-amination of electronically unbiased cyclopropanes is presented, which enables an expedient construction of a host of structurally diverse β -amino ketone derivatives. By taking advantage of a one electron oxidation, the otherwise recalcitrant donor-donor cyclopropanes are readily converted into reactive donor-acceptor cyclopropanes, which in turn readily participate in the ensuing ring-opening functionalizations." The first and second sentence of the abstract are difficult to understand and many uncommon words were used that will make it challenging for non-native readers. The authors should rephrase these sentences.

Response: As suggested by the referee, the emotional qualifiers have been deleted from the main text. The abstract has been rephrased.

Reviewer 2: 2) In the group of references 14-18, the following relevant reference should be added: *Chem.Sci.* **2018**, *9*, 6411.

Response: The relevant reference has been added in the group of reference 14-19.

Reviewer 2: 3) What is the context of references 36-48? It was unclear from the context of the sentence in which the references appeared.

Response: The sentence offering the context to references 36-48 (now is 37-49) has been reorganized as “As background, high-energy light-driven or strong oxidant-mediated ring-opening functionalization of unactivated cyclopropanes through radical cation intermediates have been reported²⁹⁻³⁶, and photoredox functionalizations of electronically activated cyclopropanes are also known³⁷⁻⁴⁹. In contrast, the visible-light photoredox catalysis approach for synthetic elaboration of relatively unreactive cyclopropanes is uncharted (Scheme 1c).”

Reviewer 2: 4) Page 2: "ring-opening and elaboration" ring-expansion?

Response: Ring-expansion relates to protocols that enable transformations from smaller rings to bigger ones. In this manuscript “ring-opening and elaboration” presents a more appropriate description of this transformation.

Reviewer 2: 5) Page 2: "Although the release of ring strain has been leveraged for functionalization reactions, a pivotal challenge is to control the selectivity of C-C bond cleavage reactions and the subsequent regiochemistry of the functionalization." Might be difficult to understand

Response: This sentence has been rephrased as “Although the release of ring strain offers a thermodynamic driving force, the control of selectivity for C–C bond cleavage and the subsequent regiochemistry of the functionalization remains difficult.”.

Reviewer 2: 6) page 2: " In stark contrast, electronically unbiased cyclopropanes prove recalcitrant toward ring-opening, much less in a regioselective fashion" Might be difficult to understand.

Response: This sentence has been rephrased as “In contrast, the ring-opening functionalization of electronically unbiased cyclopropanes is much more difficult, especially in a regioselective fashion.”.

Reviewer 2: 7) Page 2, Scheme 1: The symbol for "electron" under b) and c) should be changed from e to e⁻

Response: These changes have been made.

Reviewer 2: 8) Page 3, Table 1: "0.2 mol% PC" should be changed to "2.0 mol% PC" because these are the standard conditions used for the optimization

Response: This alteration has been made as indicated.

Reviewer 2: 9) Page 3: "~~derivatives~~ derivat~~ives~~"

Response: This typo has been revised.

Reviewer 2: 10) Page 3: "β-oxo ~~aminal~~ amino derivative **3ala**"

Response: This change has been made as indicated.

Reviewer 2: 11) Page 4, Table 2: The general structure of **1** should bear also R3 at the same carbon that R2 is connected in analogy also for the product **3**. For better understandability "**2**" should be called "**2a–2h**" and in the scheme instead of "**1**" there should be written "**1a–1ax**"

Response: We have made the suggested changes.

Reviewer 2: 12) Page 4, Fig. 1: The figure has four parts (a, b, c, d) but the description only shows a and b. This way the figure is not totally understandable. For part b the yields of **3** and **Int-3** should be added and also the yield of the conversion from **Int-3** to **3**.

Response: The descriptions of each panel in Fig. 1 have been added, and the yields of **3** and **Int-3**, as well as the yield of the conversion from **Int-3** to **3**, have been added in the Figure.

Reviewer 2: 13) Page 6, Scheme 3: For the structure of **I** it might help if one of the electrons of the drawn structure is actually removed to make it more clear, that it is a radical

and positively charged, even though the way it is drawn at the moment is not wrong. For the step from **I** to **II** "-H+" should be added below the arrow. Has the influence of additional base been investigated or is this the reason why an excess of the heterocyclic amine is used? At the step from **IV** to **3** "- H₂O" should be added beneath the arrow to make clear that it's a dehydration reaction. Product **IV** should use the same color code for the NR₂-group like before (red)

Response: The SET oxidation of aryl cyclopropane could proceed via single electron removal from the aryl group, but the positive charge is delocalized over the entire molecule. Therefore, it would be more accurate to draw the structure **I** as a general radical cation intermediate but not to specify where the oxidation happens. The "-H+" has been added below the arrow at the step from **I** to **II**, and "- H₂O" has been added beneath the arrow at the step from **IV** to **3**. The color of the NR₂ group in product **IV** has been changed to red.

Reviewer 2: 14) Page 6: "nucleophil~~e~~s"

Response: This typo has been revised.

Reviewer 2: 15) Page 7: "~~voltametry~~ voltammetry"

Response: This typo has been addressed.

Reviewer 2: 16) Page S22: Table S3: "Bule-LEDs" should be changed to "Blue-LEDs"

Response: This typo has been revised.

Reviewer 2: 17) Page S33ff: Often the frequency shown for the ¹H NMR is 401 MHz and the ¹³C NMR 101 MHz but later in the shown spectra it's always 400/100 MHz. The authors should make this consistent.

Response: This alteration has been made as indicated.

Reviewer 2: 18) For compounds **1q** page S68, **3oa** page S105 impurities are visible, cleaner spectra should be provided

Response: Improved spectra of **1q** and **3oa** have been provided in the SI (Pages 90 & 129).

Reviewer 3.

The manuscript by Walsh, Feng and co-workers details a novel method for the synthesis of beta-aminoketones from aryl cyclopropanes via a photoredox-catalysed cyclopropane ring-opening. The reaction proceeds via single electron oxidation to generate cyclopropane-substituted aryl radical cations, which undergo ring opening/nucleophilic attack by aza-heterocycles to form benzylic radicals. Subsequent reaction with molecular oxygen delivers the aryl ketone product. This work nicely builds on established reactivity

of aryl cyclopropanes, which are known to undergo single electron oxidation/nucleophilic attack/cyclopropane ring-opening to give benzylic radicals (see references 28–31, 33–35). Such reactions have previously only been performed with stoichiometric strong oxidants or with UV light sensitisation. In the present work, the authors demonstrate that the same radical intermediates can be accessed using mild photoredox conditions. Previous photoredox-catalyzed cyclopropane ring openings have been limited to easily oxidisable aminocyclopropanes and only applied to [3+2] cycloadditions with olefins (references 36–40). However, taking inspiration from the anti-Markovnikov hydroaminations of styrenes reported by Nicewicz (reference 26), the authors recognised that arylcyclopropanes could be oxidised by highly oxidising photocatalysts. The resulting radical cation intermediates show distinct reactivity from those generated from aminocyclopropanes, so could be intercepted in oxo-amination reactions to provide useful beta-amino ketone products. The reaction proceeds under mild and operationally simple conditions using easily accessible starting materials. An extensive substrate scope is demonstrated with respect to the cyclopropane, and several aza-heterocycle nucleophiles are used, effectively highlighting the generality of the methodology. Particularly attractive is the one-pot alkene amino acylation procedure. Although the yields are low, this transformation would not be possible in such a concise fashion using other methods. As a result, I think this a very interesting method that will likely be of interest to the readers of Nature Communications. I recommend publication after addressing the following points:

Response: We thank the reviewer for their supportive comments.

Reviewer 3: 1) A reference to Scheme 1d should be added to the main text.

Response: The reference for Scheme 1d has been added as ref. 66,67.

Reviewer 3: 2) In Table 1, the conditions above the reaction arrow states 0.2 mol% photocatalyst. Either this should be changed to 2.0 mol% or footnote d needs to be changed.

Response: This typo has been revised.

Reviewer 3: 3) The regioselectivity, with respect to the nucleophilic attack appears to occur through an SN-1-type process, with attack at the more substituted carbon. This would imply that reaction should be promoted by groups able to stabilise a carbocation. Therefore, the selectivity in product 3sa is intriguing as cyclopropane opening on the furan moiety would be promoted by oxocarbenium stabilisation. Is there an explanation for the observed regioselectivity? As the yield is low, is it possible that the other regioisomer is formed but the aminor product is unstable?

Response: Based on the reviewer's comments on the ring-opening functionalization of unsymmetrical diaryl cyclopropanes, the structures of **3ana** (changed to **3apa**) and **3aoa** (changed to **3aqa**) in Table 2 were reassigned based on ¹H NMR of each pair of regioisomers, and also compared with the authentic samples (**3apa**, **3apa'**, **3aqa**, **3aqa'**) made using Aza-Michael type addition.

One-pot aminoacylation of alkenes

As outlined above, we have conducted this reaction using enantioenriched diphenyl cyclopropane **(1R,2R)-1am** (changed to **(1R,2R)-1ao**). It was found that the reaction of **1am** (changed to **1ao**) of 90% ee led to the generation of the product **3ama** (changed to **3aoa**) in 65% yield with 60% ee after 24 h. This result indicates that the oxo-amination mainly proceed through a concerted nucleophilic attack/ring-opening (S_N2 -like process). The erosion of stereochemistry could arise from a minor contribution of ring-opening followed by nucleophilic attack manifold (S_N1 -like process). However, the racemization of enantiomerically enriched diphenyl cyclopropane substrate **3ama** through sensitization via triplet energy transfer, homolytic ring cleavage and radical recombination-based ring closure was also considered. To examine this possibility, the reaction of **(1R,2R)-1am** (changed to **(1R,2R)-1ao**) was monitored by chiral HPLC. Following the starting material as a function of time clearly demonstrated that the ee of the cyclopropane substrate degraded with time under the reaction conditions. Furthermore, the diastereoselective transformations for **3ata-3ava** (changed to **3ava-3axa**) are also in well accordance with concerted nucleophilic attack/ring-opening manifold. Taken together, we favor the concerted ring-opening mechanism as the predominant pathway.

The experimental results have been added in the main text (page 6, right column, lines 16-37) and related experiment details have been added in the SI (pages 43-50).

Control experiment with enantioenriched diphenyl cyclopropane

The evolution of ee values of substrate (1R,2R)-1ao and product (R)-3aoa as the reaction progresses

Diastereoselective ring-opening oxo-amination of bicyclic substrates

To further probe the intriguing result with substrate **3sa**, several control experiments were carried out. It was found that when **1ba** (1a,6b-dihydro-1H-cyclopropa[b]benzofuran) was subjected to the standard reaction conditions, only homolytic aromatic substitution product

6baa was obtained in 30% yield with no ring-opening oxo-amination observed by ^1H NMR. This phenomenon is consistent with the observation by Nicewicz that, in the case of photoredox catalyzed aromatic C–H functionalization of electron-rich arene derivatives, the nucleophile tends to attack the para-position of arenes with electron-donating substituents (*J. Am. Chem. Soc.* **2017**, *139*, 11288). Therefore, the experimental result from **1ba** indicates that after one electron oxidation the positive charge reside mainly on the aryl ring, with the cyclopropyl ring embed in oxo-bicyclic system poorly participated in charge delocalization, probably because of poor orbit overlap between the cyclopropane and radical cation. From the chart below it is seen that the bicyclic substrates **1as-1au** and **1ba**, the yield of desired product is decreased when the tether between arene and cyclopropane is shorter. Therefore, the dihedral angle between C–C bond being cleaved in cyclopropane and arene is very important for the efficiency of this reaction. The contrasting results between **1as** and **1ba** could be rationalized from the consideration of short bond length of C–O compared with C–C, resulting smaller dihedral angle in the case of **1ba**.

The reaction yield increase as the length of the tether between arene and cyclopropane grows

Through-bond delocalization SOMO ()

C–C 0.154 nm C–O 0.143 nm

Furthermore, when we submitted **1an** (1-methoxy-4-(2-methoxycyclopropyl)benzene) to the standard reaction conditions, the product **3ana** was obtained in 55% yield. In this case, the position para to the MeO- is blocked by cyclopropyl substituent. The rotational flexibility of this cyclopropyl group allows for better orbit overlap, which enables the stabilization of the positive charge from the aryl to cyclopropyl group. Thus, the N-nucleophilic attack happens preferentially on the cyclopropyl ring system. Taken together, we hypothesize that the reaction selectivity observed in the case of **3sa** is governed by the charge distribution of the radical cation intermediate.

Reviewer 3: 4) For products **3aja** and **3aka** in Table 2, it is stated that the low yield is a result of an “unfavourable inductive effect of the –CHOAc on the proximal radical cation intermediate”. This is not clear. Is it implied that the radical cation is formed and then destabilised? As this would surely help the nucleophilic attack. Or is it implied that the radical cation is not generated in the first place, due to a more difficult oxidation? If the remaining mass balance is unreacted starting material, it would suggest the latter. I was wondering whether the low yield could be a result of competing intramolecular attack of the radical cation intermediate by the carbonyl oxygen of the acyl group (for similar process, see Chem. Sci. 2015, 6, 270). Some clarification is needed.

Response: As we obtained the expected products **3aja** (changed to **3aka**) and **3aka** (changed to **3ala**), the SET oxidation of the substrates for the generation of radical cation intermediates must have taken place. As noted above, the reactions proceed predominately through a concerted nucleophilic attack/ring-opening manifold. We propose, therefore, that the positive charge of the radical cation is delocalized over the aryl and cyclopropyl moieties. Positioning an electron-donating substituent proximal to the cyclopropyl ring would affect the charge distribution, making the carbon atom of the cyclopropane less positively charged, and, therefore, less electrophilic. The cyclopropane would then be more reluctant to undergo nucleophilic attack. This description for the low yield of **3aja** (changed to **3aka**) and **3aka** (changed to **3ala**) has been added in the main text (page 4, right column, lines 22-17 from the bottom). Our previous interpretation of “unfavourable inductive effect of the –CHOAc on the proximal radical cation intermediate” has been replaced.

Furthermore, we have reexamined the reaction of **1aj** (changed to **1ak**). It was found that in addition to the formation of expected product **3aja** (changed to **3aka**), we were able to isolate a byproduct: the demethoxyamination product **5aka** (35% yield). In this case, the nucleophilic aromatic substitution by azaarene competes favorably with the ring-opening functionalization of the cyclopropane ring. We hypothesize that the proximity of the –OAc functionality slows down the ring opening, favoring the attack on the aromatic ring. The

description of the reactivity profile of **1aj** (changed to **3aka**) was added in the main text (page 3, right column, lines 25-18 from the bottom).

To further substantiate our hypothesis, two control experiments using **1bb** (ethyl 2-(4-methoxyphenyl)cyclopropane-1-carboxylate) and **1ah** (methyl 9-(2-(4-methoxyphenyl)cyclopropyl)nonanoate) were conducted. In the case of **1bb**, the reaction selectively underwent nucleophilic aromatic substitution to afford **5bba** in 50% yield, whereas the desired oxoamination product **3aha** was obtained in 45% yield when **1ah** was employed.

Related experiment results were added to the main text in Table 2 and the SI (page 41).

Reviewer 3: 5) For products **3ana**–**3apa**, regioisomeric ratios should be included

Response: Related regioisomeric ratios for **3ana-3apa** (changed to **3apa-3ara**) were added in Table 2 and the determination of these isomeric ratio by HPLC was added to the SI (Pages 50-54).

Reviewer 3: 6) What are the diastereomeric ratios of products **3ata**, **3aua** and **3ava** in Table 2? These values need to be included.

Response: The reactions with **1at**, **1au**, **1av** (changed to **1av**, **1aw**, **1ax**) were highly diastereoselective and only one diastereoisomer was obtained in each case. In these cases, the ring-opening of the cyclopropane is an S_N2 -like nucleophilic attack of the azaarene. As a result, the oxo-amination process is stereospecific. The relative stereochemistry of ring-opened products is shown in Table 2 and also in the SI.

3ava, 52%

3awa, 38%

3axa, 45%

Reviewer 3: 7) The statement “It is also worth pointing out that the azaaryl nucleophiles amenable to this reaction are, at the present stage, azaarenes with relatively high oxidation potentials” should be elaborated on and rephrased. What does “relatively high oxidation potential” mean? Relative to what? Presumably, this is to prevent competing single electron oxidation of the nitrogen nucleophile. With respect to this, have the limits of this reaction been examined? Are there examples of nucleophiles with lower oxidation potentials that have failed to react? A nice addition would be to include a brief survey of amine nucleophiles whose oxidation potentials are known/measured by CV and compared to that of the cyclopropanes. If the oxidation potential of the amine nucleophile is lower than that of the cyclopropane, does the reaction fail?

Response: We agree with the interpretation of Reviewer 3 to “prevent competing single electron oxidation of the nitrogen nucleophile” is the underlying reason for the success of aza-nucleophiles. This is what we attempted to outline in the prior version of the manuscript. The prerequisite for the success of this reaction is the efficient and selective SET oxidation of aryl cyclopropane substrates, therefore, the selected aza-nucleophile should tolerate the oxidation potentials that enable SET oxidation of cyclopropane derivatives. On the other hand, the success of the ring opening of cyclopropane radical cation intermediate is also directly affected by the nucleophilicity of azaarenes. This is the issue we faced. Nucleophiles with high nucleophilicity more easily undergo SET oxidation. As suggested by the referee, we have tested a set of aza-nucleophiles with varying oxidation potentials and found that nucleophiles with oxidation potentials lower than aryl cyclopropane substrate are not viable in the present reaction. This information has been added in the main text (page 6, left column, lines 7-15) and also included in the SI (Pages 42-43).

Aza-nucleophile	Oxidation potentials vs. SCE	Reaction result
-----------------	------------------------------	-----------------

	+0.92 V	No reaction
	+0.95 V	No reaction
	+1.15 V	No reaction
	+1.16 V	No reaction
	+1.70V	No Desired Product
	+2.21 V	Reaction succeeded
	+2.30V	No Desired Product
	> + 2.5 V	Reaction succeeded
	> + 2.5 V	aryl cyclopropane consumed but no desired product
	> + 2.5 V	aryl cyclopropane consumed but no desired product

Reviewer 3: 8) Did the authors try triflamide as the aza-nucleophile? This was used by Nicewicz (ref. 26) and provides the potential for subsequent deprotection to access primary amine products.

Response: As suggested by Reviewer 3, we examined the reaction using triflamide as nucleophile, and the reaction indeed occurred, delivering the desired product in 37% yield. This result was added in the main text (Table 2, **3ai**).

Reviewer 3: 9) The title for Figure 1 is not correct. It is missing descriptions for a and the labels for b–d are wrong.

Response: This correction has been made.

Reviewer 3: 10) In figure 1b, the yields for the conversion of Int-3ha and Int-3pa into 3ha and 3pa should be given.

Response: The yields for conversion of **Int-3ha** and **Int-3pa** into **3ha** and **3pa** have been added to figure 1b.

Reviewer 3: 11) The Stern-Volmer quenching studies should also be carried out with molecular oxygen, to rule out an oxidative quenching pathway.

Response: As suggested, the Stern-Volmer quenching study under air atmosphere was carried out and similar quenching efficiencies were observed both under N₂ and air atmosphere. Therefore, the possibility of oxidative quenching pathway by molecular oxygen could be precluded. Related experimental details were added to the SI (page 30-33).

Reviewer 3: 12) For ease of comparison, I think the Stern-Volmer plots would be better presented on a single graph.

Response: As suggested, this change has been made.

Stern–Volmer fluorescence quenching study of 1a & 2a

Reviewer 3: 13) A reference is needed for König's work (ref. 44) during the discussion of the mechanism in Scheme 3.

Response: This reference has been added.

Reviewer 3: 14) In the discussion it is stated “Attack occurs at the more substituted carbon of the cyclopropyl radical cation, likely due to the greater partial positive charge at this position. Consistent with this proposal, under standard conditions the reaction of the unsymmetric *trans*-1,2-diarylcyclopropane bearing both a 4-OMe substituent on one aryl and a 4-CF₃ on the other results in selective formation of the keto group adjacent to the electron donating 4-C₆H₄-OMe group.” These two regioselectivity arguments are not related. The first relates to the regioselectivity of the cyclopropane ring opening (quaternary position vs. secondary position) and the second relates to regioselectivity of single electron oxidation (electron-rich aromatic vs. electron deficient aromatic).

Response: The interpretation of reaction regioselectivity has been revised. “This regioselectivity of nucleophilic attack on aryl cyclopropyl radical cation intermediate is

affected by the electronic effects of substituents on the cyclopropyl ring and selectively occurs where the partial positive charge is more readily accommodated. In the case of mono-aryl cyclopropanes, attack occurs at the most substituted carbon, which bears greater partial positive charge. For unsymmetric diary cyclopropanes, nucleophilic attack takes place preferentially at the cyclopropane carbon with the more electron-donating aryl group.

Reviewer 3: 15) Add a Table of contents.

Response: Table of contents has been added in the SI.

Reviewer 3: 16) The diastereomeric ratios of products 3ata, 3aua and 3ava need to be given. The NMR spectra suggest a single diastereomer. The major diastereomer should be assigned and the relative stereochemistry included in both the supporting information and the manuscript.

Response: The reactions with **1at**, **1au**, **1av** (changed to **1av**, **1aw**, **1ax**) were stereospecific processes, and only one diastereoisomer was obtained. In these cases, the ring-opening of the cyclopropane is induced by the nucleophilic attack of azaarene and, therefore, the oxo-amination process is stereospecific, resembling a concerted S_N2 reaction mechanism. The relative stereochemistry of products is illustrated in Table 2 and also in the SI.

Reviewer 3: 17) For the photon flux measurement, two different values are given: 7.41×10^{-7} and 4.27×10^{-7} einstein/s. Which is correct?

Response: The photon flux was re-measured with value of 2.351×10^{-8} einstein/s.

Reviewer 3: 18) The photon flux calculation should be repeated with more time points at less than 90 s to check the accuracy of the value. Based on the numbers given, 4.27×10^{-7} moles of Fe^{2+} are generated, however, only 0.35 mL of a 5.55 mM phenanthroline solution is added, which allows for a maximum of only 6.48×10^{-7} moles of $\text{Fe}(\text{phen})_3^{2+}$. Therefore, it is likely that all the phenanthroline has been consumed, which will result an incorrect photon flux value.

Response: The quantum yield was recalculated, and the details for the measurement was added in the SI (page 34-39).

Determination of the Reaction Quantum Yield (Φ)

Emission spectrum of blue LED used for quantum yield experiments ($\lambda_{max} = 459$ nm). Recorded using a F-4600 FL Spectrophotometer.

Determination of the Light Intensity at 459 nm

Following a modified procedure reported by Melchiorre and co-workers,¹⁶ an aq. ferrioxalate actinometer solution was prepared and stored in the dark. The actinometer solution measures the photodecomposition of ferric oxalate anions to ferrous oxalate anions, which are then reacted with 1,10-phenanthroline to form $\text{Fe}(\text{Phen})_3^{2+}$. Its concentration is then estimated by UV/Vis absorbance at 510 nm. The number of moles of $\text{Fe}(\text{Phen})_3^{2+}$ complex formed is related to the numbers of photons absorbed by the actinometer solution. Preparation of the solutions used for the studies:

1. Potassium ferrioxalate solution: Potassium ferrioxalate trihydrate (118 mg) and 95-98% H_2SO_4 (56 μL) were added to a 20 mL volumetric flask and filled to the mark with distilled water.
2. Buffer solution: Sodium acetate (0.988 g) and 95-98% H_2SO_4 (0.2 mL) were added to a 20 mL volumetric flask and filled to the mark with distilled water.

The actinometry measurements:

- a) 1 mL of the actinometer solution was taken in a quartz cuvette ($l = 1$ cm). Both the cuvettes of actinometer solution and reaction solution were placed next to each other at a distance of 5 cm away from a 15 W blue LED ($\lambda_{\text{max}} = 459$ nm) and irradiated for 30 s. The same process was repeated for different time intervals: 60 and 90 s.
- b) After irradiation, the actinometer solution was transferred to a 10 mL volumetric flask containing 1.0 mg of 1,10-phenanthroline in 2 ml of buffer solution. The flask was filled to the mark with distilled water. In a similar manner, a blank solution (10 mL) was also prepared using the actinometer solution stored in dark.
- c) Absorbance of the actinometer solution after complexation with 1,10-phenanthroline at $\lambda = 510$ nm was measured by UV/Vis spectrophotometry.
- d) According to Beer's law, the number of moles of Fe^{2+} formed (x) for each sample was determined by:

$$\text{mol Fe}^{2+} = \frac{v_1 \cdot v_3 \cdot \Delta A(510\text{nm})}{1000 \cdot v_2 \cdot l \cdot \epsilon(510\text{nm})}$$

Where:

v_1 = Irradiated volume (1 mL).

v_2 = The aliquot of the irradiated solution taken for the estimation of Fe^{2+} ions (1 mL). v_3

= Final volume of the solution after complexation with 1,10-phenanthroline (10 mL). $\epsilon(510$ nm) = Molar extinction coefficient of $[\text{Fe}(\text{Phen})_3]^{2+}$ complex ($11100 \text{ L mol}^{-1}\text{cm}^{-1}$).

l = Optical path-length of the cuvette (1 cm).

$\Delta A(510 \text{ nm})$ = Difference in absorbance between the irradiated solution and the solution stored in dark (blank).

e) The number of moles of Fe^{2+} formed (x) was plotted as a function of time (t). The slope (dx/dt) of the line is equal to the number of moles of Fe^{2+} formed per unit time.

f) This slope (dx/dt) was correlated to the number of moles of incident photons per unit time (F = photon flux) by using following equation:

$$\Phi(\lambda) = \frac{\frac{dx}{dt}}{F \cdot (1 - 10^{-A(\lambda)})}$$

$\Phi(\lambda)$ = The quantum yield for Fe^{2+} formation at 450 nm is 0.9.¹⁷

g) $A(\lambda)$ = Absorbance of the ferrioxalate actinometer solution at a wavelength of 459 nm, which was measured placing 1 mL of the solution in a cuvette of pathlength 1 cm by UV/Vis spectrophotometry.

Sample calculation:

$$A^0 = 0.023, \quad A^{1_{30s}} = 0.772, \quad A^{1_{60s}} = 1.319, \quad A^{1_{90s}} = 1.918$$

$$\Delta A^{1_{30s}} = 0.749 \quad \Delta A^{1_{60s}} = 1.296 \quad \Delta A^{1_{90s}} = 1.895$$

$$\begin{aligned} \text{mol Fe}^{2+} (30 \text{ s}) &= (1 \text{ mL} \times 10 \text{ mL} \times 0.749) / (1000 \times 1 \text{ mL} \times 1 \text{ cm} \times 111100 \text{ L mol}^{-1} \text{ cm}^{-1}) \\ &= 6.748 \times 10^{-7} \text{ mol} \end{aligned}$$

$$\begin{aligned} \text{mol Fe}^{2+} (60 \text{ s}) &= (1 \text{ mL} \times 10 \text{ mL} \times 1.296) / (1000 \times 1 \text{ mL} \times 1 \text{ cm} \times 111100 \text{ L mol}^{-1} \text{ cm}^{-1}) \\ &= 1.1676 \times 10^{-6} \text{ mol} \end{aligned}$$

$$\begin{aligned} \text{mol Fe}^{2+} (90 \text{ s}) &= (1 \text{ mL} \times 10 \text{ mL} \times 1.895) / (1000 \times 1 \text{ mL} \times 1 \text{ cm} \times 111100 \text{ L mol}^{-1} \text{ cm}^{-1}) \\ &= 1.7072 \times 10^{-6} \text{ mol} \end{aligned}$$

Moles of $[\text{Fe}(\text{Phen})_3]^{2+}$ per unit of time formed due to decomposition of the actinometer solution at 459 nm blue Led irradiation

$$F = \frac{\frac{dx}{dt}}{\Phi(\lambda) \cdot (1 - 10^{-A(\lambda)})}$$

$$A(\lambda)_{30s} = 0.649, \quad A(\lambda)_{60s} = 1.134, \quad A(\lambda)_{90s} = 1.638$$

$$F_{30s} = (1.8715 \times 10^{-8}) / (0.9 \times (1 - 10^{-0.649})) = 2.681 \times 10^{-8}$$

$$F_{60s} = (1.8715 \times 10^{-8}) / (0.9 \times (1 - 10^{-1.134})) = 2.244 \times 10^{-8}$$

$$F_{90s} = (1.8715 \times 10^{-8}) / (0.9 \times (1 - 10^{-1.638})) = 2.128 \times 10^{-8}$$

$$F_{\text{average}} = (F_{30s} + F_{60s} + F_{90s}) / 3 = 2.351 \times 10^{-8}$$

h) The determined incident photons per unit time (F) is 2.351×10^{-8} einsteins/s.

Absorbance of the ferrioxalate solution.

Determination of the Reaction Quantum Yield

To a 3 mL quartz cuvette with two sides taped over with electrical tape, **1a** (35 mg, 0.2 mmol, 1 equiv), **2a** (27 mg, 0.4 mmol, 2 equiv), [Ir(dF(CF₃)ppy)₂(4,4'-bpy)](PF₆) (PC-III) (1.0 mg, 0.2 mol%) and anhydrous DCE (1.0 mL, 0.2 M), 4Å-MS (100 mg) and a small stir bar were added and then the quartz cuvette was capped and charged with O₂ using a balloon. The sample was stirred and irradiated for 10800 s (3.0 h) at $\lambda_{\text{max}} = 459 \text{ nm}$ at rt. After irradiation, the yield of product **3aa** was determined to be 16.6% (3.32×10^{-5} mol of **3aa**) by ¹H NMR integration against an internal standard. The reaction quantum yield (Φ) was determined using the formula below where the photon flux is 2.35×10^{-8} einstein s⁻¹ (described above), t is the reaction time (10800 s) and f is the fraction of incident light absorbed by the reaction mixture. An absorbance of the reaction mixture at 459 nm was measured to be 1.426

$$\Phi = \frac{\text{mol of product formed}}{\text{photon flux} \cdot t \cdot f}$$

Sample quantum yield calculation

$$f = 1 - 10^{-1.426} = 0.9625$$

$$\Phi = 3.32 \times 10^{-5} \text{ mol} / (2.351 \times 10^{-8} \text{ einstein s}^{-1} \times 10800 \text{ s} \times 0.9625) = 0.14$$

The reaction quantum yield (Φ) was thus determined to be 0.14.

Absorbance of the reaction mixture solution.

Reviewer 3: 19) There is a problem with the references. E.g. for the quantum yield measurements, references 13–17 are given, however, only 15 references are given in the Supporting Information.

Response: The omitted references have been added in the SI.

Reviewer 3: 20) The NMR of products **3ada** (p. S121), **3aea** (p. S122), **3ara** (p. S135) and **3asa** (p. S136) have significant impurities present. The products should be re-purified, and the yield recalculated.

Response: We have re-conducted these four experiments, the yields of isolated were recalculated and added in Table 2 and the NMR spectra were added in SI. **3ada** (p. S145), **3aea** (p. S146), **3ara** (changed to **3ata**, p. S164) and **3asa** (changed to **3aua**, p. S165)

REVIEWERS' COMMENTS:

Reviewer #1 (Remarks to the Author):

Based on the point by point response and the revised manuscript, all the points I raised are solved. the additional mechanistic studies and discussions are convincing. I have no further comments and suggest to publish

Reviewer #2 (Remarks to the Author):

The authors have done a remarkable job of addressing the concerns of the referees. I can recommend publication at this stage. Congrats on a very nice manuscript!

Reviewer #3 (Remarks to the Author):

The authors have given detailed responses to all of the comments and made appropriate changes to the manuscript and SI. I am happy to recommend the manuscript for publication.

Adam Noble